# Training open-ended policies to follow video-prompt instructions with reinforcement learning

## Abstract

In recent years, online reinforcement learning(RL) training methods like PPO have shone in important works such as Instruct GPT. However, unlike the success achieved in the language domain, online RL methods often struggle to generalize to untrained tasks in open-world environments like Minecraft, due to issues like overfitting. This has become a significant obstacle in using online methods to build a generalist agent. In this work, we notice the modality differences between natural language environments and embodied environments such as the Minecraft environment, which inspired us to use video instructions instead of text instructions to enhance the model's understanding of the relationship between the environment and instructions. We also introduce a new attention layer in the base model's encoder-decoder architecture to establish a semantic and visual dual-path information interaction channel, further strengthening this generalization capability. After training our model on a small set of tasks, it demonstrated excellent zero-shot generalization on new tasks, outperforming almost all other models in the Minecraft environment on our benchmark. Our approach takes a solid and important step toward unleashing the potential of online RL in building generalist agents.

## 1 Introduction

In recent years, with the rising prominence of methods like Reinforcement Learning with Human Feedback (RLHFOuyang et al. (2022b)) and InstructGPTOuyang et al. (2022a), online reinforcement learning(RL) methods such as Proximal Policy Optimization (PPOSchulman et al. (2017)) have attracted increased attention from researchers by its strong performance. However, traditional online RL is often used only for a single task and often suffers from overfitting. With the rise of the concept of generalist agents, applying online RL to instruction-following agents and achieving generalization abilities in open-ended tasks has become a challenge. To seek a solution to this problem, we try to combine online RL with multi-task training to harness their potential in building general intelligence. In this procedure, we noticed that the ability to generalize from training on a few tasks to solving open-ended tasks is the key. Existing work primarily focuses on text-based instructions, which limits the model's ability to generalize to new tasks. On the other hand, language-visual alignment in traditional text instruction methods requires collecting a large amount of text and trajectory samples during the text-based pre-training and the online tasks defining, where the text is significantly more expensive than the trajectories.

In this work. We draw inspiration from recent works on video instructionsCai et al. (2023) and explore the way to leverage video instructions and jointly online finetunes an encoder-decoder structure. This approach, compared to using traditional text instructions, ensures that the agent's input instructions and the actual running environment are in the same modality, naturally eliminating the complex and challenging multimodal alignment process inherent in traditional text instruction methods, enabling the agent to more intuitively and easily connect instructions to policies. Our work demonstrates the zero-shot generalization capability of online RL with video instruction and highlights the potential of online RL in building open-ended generalist agents.

Figure 1: This figure provides a high-level summary of the contribution made by Video Instruction to generalization. In addition to obtaining a large amount of inexpensive data, Video Instruction eliminates the text-video alignment step found in traditional methods, allowing the agent to more easily associate the common latent features between the observation and the instruction. This simplicity also enables the encoding and decoding of these common latent features to generalize to unseen instructions.

In practice, we found that directly using video instructions to perform online finetuning on the encoder-decoder structure of the base model results in noticeable generalization performance, even when the number of tasks is limited. However, there is still a tendency for the latent $z$ encoded by the instruction to overfit the training task space, which restricts its generalization ability in the unseen task space. We believe that during the model's pre-training process, the latent is shaped primarily for conveying action features. Therefore, when we perform online finetuning on a small number of tasks that demand a high level of understanding of visual features, the model may maintain a latent vocabulary to encode and decode the corresponding visual features and processing patterns. This introduces difficulties in generalizing to new tasks. To address this issue, we propose a dual-pathway encoder-decoder model architecture that processes both semantic and visual features, helping the model find the relationship between the observation and instructions and utilize the semantic features to decode the latent. This structure indicates that naturally and succinctly guiding the model to find the connection between instructions and observations is key to advancing the generalization capability of online algorithms, and significantly improving the model's generalization ability.

The remaining part of the chapter will provide a more detailed description of our work. In Chapter 2, we will focus on the challenges currently faced in building general agents, highlighting our motivation and the significance of this work. Chapter 3 will detail our approach, including the introduction of the video instruction structure and the construction of a dual-path encoder-decoder model architecture. In Chapter 4, we will present our main experimental results and some ablation studies, including comparisons of our model's capabilities in the Minecraft environment against other online methods, showcasing the impressive zero-shot generalization ability of online RL. Finally, we will discuss the potential of integrating online methods into open-ended task environments, illustrating how training on a small number of tasks can contribute to constructing general intelligence.

## 2 PRELIMILARY

### 2.1 PROBLEM FORMULATION

We formulate a goal-conditioned task in Minecraft environment as a finite-horizon Markov Decision Process(MDP) $\langle \mathcal{S}, \mathcal{A}, \mathcal{R}, \mathcal{P}, d_0, g \rangle$, where $\mathcal{S}$ is the state space, $\mathcal{A}$ is the action space, $\mathcal{R} : \mathcal{S} \times \mathcal{A} \to \mathbb{R}$ is the reward function condition on the goal g. $\mathcal{P} : \mathcal{S} \times \mathcal{A} \to \mathcal{S}$ is the transition dynamics, $d_0$ is the initial state distribution. The optimization's goal is to learn a policy $\pi(a|s, g)$ that maximizes the expected cumulative reward $\mathbb{E}\left[\sum_{t=0}^{\infty} \gamma^t r_t^g\right]$, where $\gamma \in (0, 1]$ is a discount factor and $r_t^g$ is the reward on the t-th step under policy g in an episode.

## 2.2 MULTI-TASK REINFORCEMENT LEARNING IN MINECRAFT ENVIRONMENT

Multi-task reinforcement learning (multi-task RL) involves training a single agent to perform multiple tasks simultaneously, with the goal of leveraging shared knowledge across tasks to improve learning efficiency and performance. Instead of training separate models for each task, multi-task RL aims to develop a unified policy that can generalize across different environments or objectives. Moreover, in environments with open-ended tasks, such as Minecraft, multi-task RL can only train on a limited set of tasks. Therefore, the agent must also possess the ability to generalize in a zero-shot manner to new, unseen tasks based on the knowledge gained from existing tasks.

Overall, multi-task online reinforcement learning in Minecraft faces several main challenges:

**Task Interference**: Learning multiple tasks can lead to negative transfer, where optimizing for one task may hinder performance on others. Also, different tasks may require different strategies, making it difficult to balance exploration and exploitation across tasks.

**Generalization Difficulties**: Online fine-tuning on a small set of tasks can lead to overfitting on easy tasks that provide relatively higher rewards, which may in turn compromise the generalization capability of the base model. Even if the model performs well on most tasks in the training set, it can still easily encounter out-of-domain situations in open-ended environments, leading to failure in test tasks.

**Complexity of Environment**: Due to the difficulties like task variety, sparce reward, and dynamic environments, Minecraft becomes an extremely complex and challenging environment for multi-tasking RL. Therefore we choose Minecraft as our test environment to highlight the effectiveness of our approach. More details of this part will be covered in the appendix.

## 3 METHOD

### 3.1 VIDEO INSTRUCTED PRE-TRAINING

Several base agents in the Minecraft environment are worth noticing: VPT(Baker et al. (2022)) leveraged large-scale human-annotated data and model-generated labels to perform offline training on the transformer architecture, achieving strong capabilities. Through subsequent online fine-tuning, it became the first agent to mine diamonds using basic keyboard and mouse operations. However, VPT could not follow human instructions. Attracted by VPT's strong potential for instruction tuning, STEVE1(Lifshitz et al. (2023)), which can follow text instructions, and GROOT(Cai et al. (2023)), which can follow video instructions, were trained, and becoming models in the Minecraft environment that possess considerable instruction-following abilities.

Previous Minecraft works, such as STEVE-1 and MineCLIP agent, have focused on more traditional instruction-following agents, which often use textual instructions as input. The typical approach involves fine-tuning models based on the CLIP architecture in the Minecraft environment, using the model to encode the textual instructions. These encoded instructions are then integrated into the network's processing of observations to generate the output actions. However, In the Minecraft environment, there are often a large number of visual and semantic features. Many visual details, such as item thumbnails in the inventory, objects partially visible on the screen, and items with only subtle visual differences, require the model to have a thorough understanding of the fine details in the images. Additionally, encoding the 3D environment of Minecraft through video requires the visual encoding model to have the capability to associate the same object from different angles of view and embed it in a way that aligns with the corresponding text in a similar embedding space. These factors pose significant challenges for video-text aligning models like CLIP, as they must accurately capture and link multi-view visual representations with their textual counterparts in a complex and dynamic environment. Additionally, most of the training data for aligning models like CLIP is collected from the internet, where semantic ambiguity is prevalent. As a result, when a player is holding an axe among a herd of cows and sheep, CLIP struggles to differentiate whether the scene represents 'killing sheep,' 'killing cows,' or simply 'wandering around,' despite the significant semantic differences between these texts. Meanwhile, the videos may appear highly similar. These factors make cross-modal alignment even more challenging.

These difficulties directly lead to models heavily relying on prior experiences and memory to associate familiar objects from different viewpoints with their corresponding text descriptions, as well as linking specific behavior patterns and visual features—like tools in hand—with common action texts. When presented with rare or entirely new objects, actions, or unseen video-text combinations, visual-language multimodal alignment models like CLIP often face serious generalization issues. This reliance on familiar patterns weakens their ability to handle novel inputs effectively.

With an understanding of the alignment challenges faced by text instructions, we opted to explore using video instructions to enhance the model's generalization ability, which became a natural choice. The advantage of video instructions over text instructions is that the model can directly associate visual features without requiring a complex cross-modal alignment process which may hurt diversity. This allows for a more intuitive connection such as the common visual features between the instructions and the visual environment. Therefore it may improve the model's ability to generalize to new tasks and scenarios.

Given GROOT's outstanding ability to follow video instructions, we chose it as our base policy. The main idea behind GROOT is to use an encoder-decoder structure. The encoder encodes the input video instructions into a latent representation, which is then combined with the environment's observation. The decoder uses this latent representation along with the observation to generate the corresponding actions. More detail about GROOT as our base policy is covered in section 3.3.

## 3.2 KL-Constrained Reinforcement Learning

As one of the most famous and commonly used online RL algorithms, Proximal Policy Optimization (PPO)Schulman et al. (2017) is the primary online reinforcement learning algorithm we use in this work. In each iteration, we optimize KL-Constrained PPO loss:

$$L^{CLIP}(\theta) = \mathbb{E}_t \left[ \min \left( r_t(\theta)\hat{A}_t, \text{clip}(r_t(\theta), 1 - \epsilon, 1 + \epsilon)\hat{A}_t \right) + \lambda KL(\pi_\theta(a_t|s_t), \pi_{\theta_{\text{sft}}}(a_t|s_t)) \right]$$

where $r_t(\theta) = \frac{\pi_\theta(a_t|s_t)}{\pi_{\theta_{\text{old}}}(a_t|s_t)}$ is the probability ratio, $\hat{A}_t$ is the estimated advantage:

$$\hat{A}_t = \sum_{l=t}^{t+T} (\gamma\lambda)^l (r_l + \gamma V_\theta(s_{l+1}) - V_\theta(s_l))$$

at time $t$, T is the time stamp we set to estimate the advantage. $r_t$ is the environment reward in time t, $\gamma$ is the discount factor, and $\lambda$ is a parameter that balances bias and variance in the advantage estimates. $V_\theta(s_l)$ represents the value function and estimates the expected discounted reward from state s and is computed by a neural network that is trained simultaneously. $\epsilon$ is a small hyperparameter that defines the clipping range.

Furthermore, we introduce an additional KL term between the current policy and the original policy. This term helps prevent catastrophic forgetting in the agent and ensures that it can obtain sufficient rewards during the early stages of reinforcement learning. Its coefficient $\lambda$ will decay as the number of training epochs increases.

## 3.3 Intention aware Attention

In practical experiments, although the online fine-tuning of the agent GROOT with video instructions demonstrated some generalization ability, it was not as remarkable as expected. After analysis, we believe this reflects inherent issues within the encoder-decoder structure: Since GROOT's pre-training is based on imitation learning using a large amount of contractor data collected in the VPT work, In such a setup, the latent $z$ that used by decoder to predict action tends to focus more on encoding the tools in hand and the behavioral logic shown in the current video to predict actions, rather than focusing on the detailed information of the objects in view.

Once such a pattern is established, it becomes difficult for online fine-tuning to revise the entire internal logic of the pipeline. When we online finetune the base policy on tasks that require more awareness of visual features, to ensure that the model can perform normal initial exploration, maintain basic behaviors of the base agent, and limit the KL constraint between the current agent and the base agent in our optimization procedure, the majority of dimensions in $z$ still pass semantic

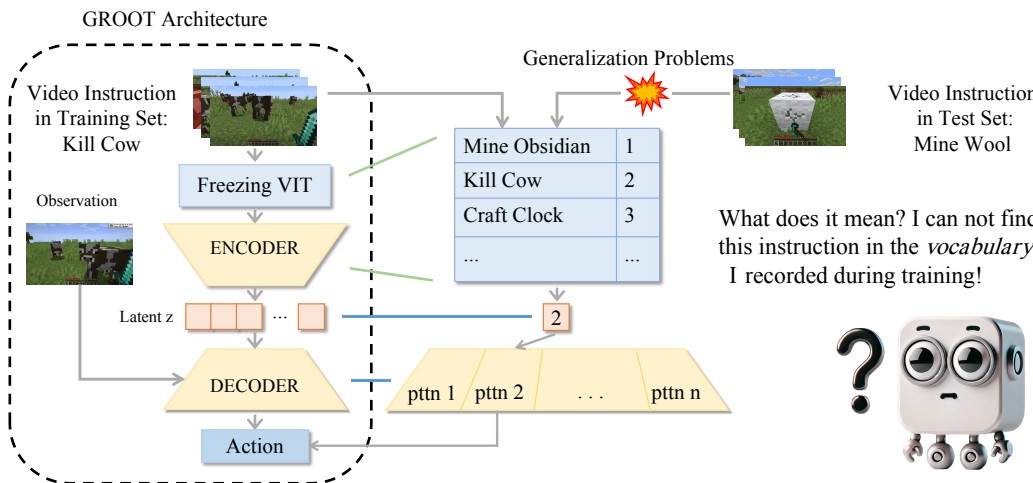

Figure 2: **Base Model's Architecture and Generalization Problems:** This figure illustrates the conventional encoder-decoder architecture of our base model and the generalization challenges it faces during online training: the latent $z$ tends to encode actions during pre-training. This may lead to $z$ relying on a limited set of feature dimensions to maintain an instruction-latent vocabulary during online finetuning, making it difficult to generalize to unseen data.

information. However, since the dimension of $z$ is relatively small and there is a higher demand for transmitting fine-grained visual features during tasks than for pre-training, the agent may encode different visual features into a built-in latent vocabulary, effectively *memorizing* the task environment. For example, the last dimension of the latent vector might represent the task environment: a value of `1` could indicate `Mine Obsidian`, `2` for `Kill Cow`, `3` for `Craft Clock`, and so on. The decoder then reads the last dimension of the latent vector and decodes this vocabulary, selecting one of several different behavior policies learned during training, each tailored for a specific task. This strategy allows the agent to achieve very high rewards for tasks in the training set. However, when encountering a completely new environment, the agent is unable to encode the unseen environment into the latent vocabulary, leading to a lack of generalization ability.

To avoid the generalization issue caused by this "task memorization", we consider leveraging the advantages of video instructions. While minimally impacting the performance of the base agent, we aim to introduce a new intention-aware pathway that directly transfers visual features before encoding observation feature patches into the latent $z$. Since the core issue lies in the fact that the model's latent $z$ was shaped during pre-training to primarily convey semantic information that is more beneficial for action derivation, this results in most of $z$'s dimensions being used to transmit semantic information, leaving too few degrees of freedom to transmit visual information. During online fine-tuning, we hope the model can leverage this new pathway, which provides more degrees of freedom and a more direct route. This would allow the successful transmission of visual information from the instructions without significantly affecting the transmission of semantic information in the original $z$.

The specific structure we designed is shown in Figure 2, we additionally introduce an intention-aware cross-attention layer within the framework to help the model better connect visual features. During the rollout process, video instructions and the current observation are passed through a freezing ViT encoder to obtain image feature patches. The latent $z$ obtained from encoding the observation is also processed through a linear layer to produce a patch. The patches from the observation are used as the query key, and after performing cross-attention with the keys and values derived from the other patches through a linear layer, we obtain the embeddings for each patch in the observation. These embeddings are then added to the corresponding patches in the original observation. The resulting patches are subsequently sent to the encoder for encoding and decoding operations in the original workflow.

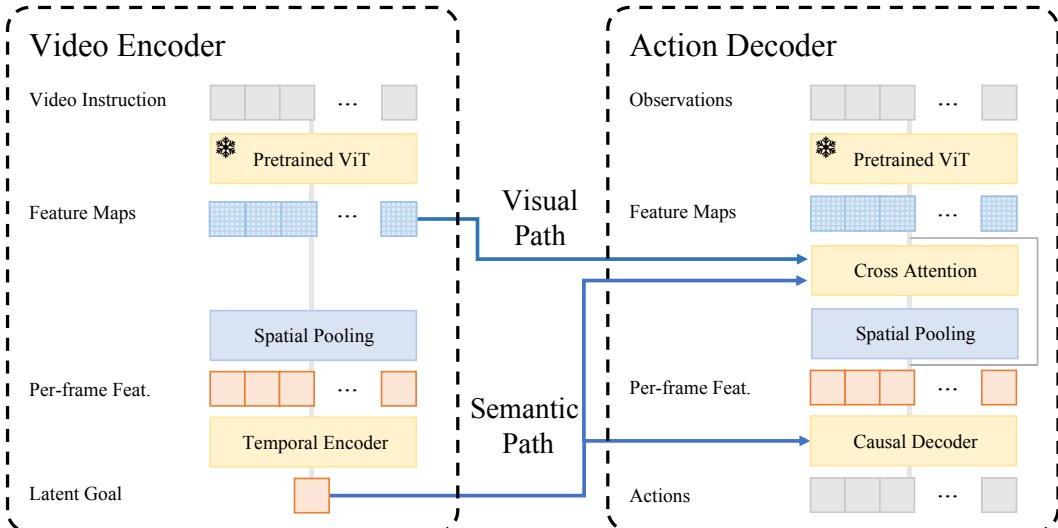

Figure 3: Draft: The attention and dual-pathway model architecture)

This structure, with awareness of the latent $z$, conveys the image-matching information between the observation and instruction through cross-attention. This forms a dual-pathway structure of high-level semantics and low-level vision: (1) The semantic pathway, encoded by the reference-conditioned policy pre-trained on unlabeled data, provides high-level behavioral information. This information can effectively enhance the exploration process in RL and exert macro-level control over the model's behavior. (2) A visual communication module based on a semantics-aware attention mechanism provides more fine-grained visual features for the specified task, allowing the policy to directly focus on the relevant visual concepts within the reference video.

Since we are essentially only altering the image input that the encoder accepts, during the initial exploration phase of online fine-tuning, the base agent can still utilize the semantic and action information conveyed by the latent $z$ to process slightly perturbed observation feature encodings, without significantly affecting performance. This dual-path architecture allows the agent to retain the ability to complete tasks during the initial exploration phase of online training, thereby achieving a stable initial reward that supports basic exploration and training.

## 4 EXPERIMENTS

### 4.1 EXPERIMENTAL SETUP

We select the open-ended video game Minecraft as the testbed (Fan et al., 2022; Guss et al., 2019). We primarily designed experiments for three categories of tasks to test the model's generalization ability from various perspectives: `MineBlock`, `CraftItem` and `KillEntity`, which are the basic skills in an open-ended Minecraft environment. The agent is required to follow instructions to mine the correct blocks, craft the accurate items, and kill the corresponding types of animals respectively in these three categories of tasks. We recorded each task's instruction videos manually and generated them using the model, showcasing the player successfully performing the correct actions on the target objects. To align with the actual instructional scenarios used during the design of the agent, the instruction videos do not include any distractors which we will mention later. We aim for the agent to perform the corresponding action on the corresponding items shown in the instruction video. For example, if the instruct video shows the player crafting a clock, then the agent's task is to craft a clock as well. For each task, when the agent performs the correct and complete action on the appropriate item—such as mining, killing, or crafting—it receives the corresponding reward. In addition to the correct items, we introduce other items as distractors. When the agent performs actions on these distractor items, we correspondingly reduce the reward as the punishment. For example, in task `mine obsidian` in the train dataset of `MineBlock` category, We will use commands to spawn the agent on a plain, then randomly select three positions on the

ground within an 11x11 area centered around the agent, excluding a 2x2 area. One position will be filled with an obsidian block, and the other two positions will be filled with distractor blocks from the training set, such as `white wool` and `oak log`. In this setting, if the model successfully mines the obsidian, it will receive a reward; if it mines the distractor blocks `white wool` or `oak log`, it will lose a small portion of the reward. This setting reasonably encourages the model to explore and correctly mine the target. More specific details about the task setting are provided in the appendix.

To accurately measure the agent's performance and generalization ability from multiple dimensions, we collected a different number of corresponding entities for each task: 160 blocks for `Mine Block`, 64 items for `CraftItem`, and 13 entities for `KillEntity`. Each of these three types of tasks has a different inclination in terms of completion and generalization difficulty. In the `KillEntity` task, there are fewer types of entities, often involving observations from various angles and the spontaneous movement of the same object on the map, testing the model's ability to generalize from a small number of training samples and match features across different views of the same object. The `CraftItem` task requires the model to focus on the thumbnail of the item, presenting a high level of difficulty in visual feature extraction. Additionally, we provided the model with some entirely unrelated initial materials, meaning there are numerous distractors in the crafting options (often more than 15, compared to only 2 in the other categories). The `MineBlock` task has the largest variety of entities and a more balanced setup in terms of distractors and visual difficulty. The model must understand subtle differences between various entities and generalize across a wide range of others. Among these task setups, `CraftItem` is the most challenging, followed by `KillEntity`, with `MineBlock` being the easiest. However, due to the diversity of entities, some very rare entities are also included, which means that even the simplest task, `MineBlock`, can present significant challenges for the agent.

## 4.2 MAIN RESULTS

Table 1 shows our main results. These results were measured over 500-1000 rollouts in unseen settings. The left part roughly reflects the agent's accuracy, i.e., how well the target task was completed, while the right part reflects the agent's precision, i.e., the exactness with which the task was completed. In this table, we compare our agent to several baselines: (a)VPT(Baker et al. (2022)), a base model obtained by pre-training on a large dataset labeled using an inverse dynamic model and imitation learning; (b)STEVE-1(Lifshitz et al. (2023)), a model built on VPT that utilizes the finetuned CLIP architecture MineCLIP(Fan et al. (2022)) to achieve text instruction following capability; (c) GROOT, a model with an encoder-decoder architecture based on causal transformers, capable of following video instructions. These comparison results were measured over 500-1000 rollouts in unseen settings The left part roughly reflects the agent's accuracy, i.e., how well the target task was completed, while the right part reflects the agent's precision, i.e., the exactness with which the task was completed.

It can be observed that after finetuning with the training dataset, our method also achieved impressive results on the test dataset and almost outperformed all other agents in every task. This demonstrates the strong generalization capability exhibited by video instruction online fine-tuning on the base model. Another point to note is that in our statistics, even if a distractor is mined during a rollout, as long as the agent successfully mines the correct block, it is still counted towards the success rate. Therefore, the success rate and accuracy are not necessarily positively correlated; in fact, they can be negatively correlated. An aggressive agent that interacts more frequently with objects may achieve a higher success rate due to the increased number of interactions, but its accuracy may be lower. Conversely, a more conservative agent that interacts with objects only when it is confident may have a relatively lower success rate but higher accuracy. For instance, STEVE1 has good accuracy in the crafting task, but its success rate is not high. This probably indicates that STEVE1 tends to complete simpler tasks that it excels at, thereby ensuring higher accuracy. Therefore, it is essential to consider these statistics comprehensively. In fact, our model often outperforms the baselines in both success rate and accuracy, demonstrating its excellent generalization capability.

Fig 4.2 presents the results of our agent and the baseline on several randomly selected tasks, covering both in-distribution and out-of-distribution scenarios. In this experiment, we can also observe the performance differences of the same model when facing different tasks. For example, Groot's ability to craft an iron pickaxe is significantly better than its ability to craft a clock. However, human players

Table 1: **Generalizaiton Test:** The main result in three categories of tasks on *unseen tasks*. The values of the left and right halves represent the agent's success rate on all rollout segments and the agent's accuracy in interactions with all objects including targets and distractors, respectively.

| Method | Success Rate(%) | | | Accuracy(%) | | |
|---|---|---|---|---|---|---|
| | Mine Block (160, 60) | Craft Item (64, 21) | Kill Entity (13, 6) | Mine Block (160, 60) | Craft Item (64, 21) | Kill Entity (13, 6) |
| VPT (Baker et al., 2022) | 2.0 | 0.4 | 6.2 | 27.4 | 10.5 | 42.5 |
| STEVE-1 (Lifshitz et al., 2023) | 3.0 | 7.4 | 30.8 | 27.8 | **15.8** | 47.2 |
| GROOT (Cai et al., 2023) | 16.5 | 1.0 | 29.7 | 35.2 | 8.2 | 47.0 |
| Ours | **69.1** | **16.0** | **67.7** | **70.8** | 12.3 | **77.2** |

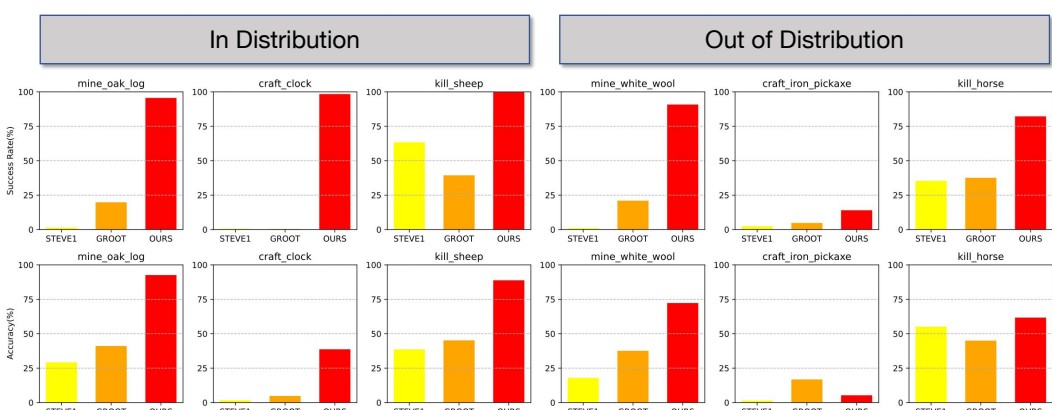

Figure 4: **Task specified Results:** We selected several tasks commonly encountered in the Minecraft environment to test our model. It can be observed that even for tasks frequently involved in pre-training, our agent outperforms other agents in both in-distribution and out-of-distribution tests. This demonstrates that our online method provides the agent with excellent generalization capabilities.

generally consider the difficulty of these two tasks to be the same. The reason for this discrepancy lies partly in the fact that crafting an iron pickaxe is a classic task in the Minecraft environment, with a large amount of related data present in Groot's pre-training dataset, while data for crafting a clock is relatively scarce. On the other hand, during GROOT's pre-training, there was no effective generalization of tasks within the same category; it did not generalize the ability to craft a clock from the abundant learning of crafting an iron pickaxe. Our agent, however, can effectively address this issue: although there is still a gap in the agent's capabilities between in-distribution and out-of-distribution tasks, it can still achieve excellent performance on out-of-distribution tasks.

### 4.3 ABLATION ON INTENTION-AWARE ATTENTION

Figure 4.3 shows our ablation experiments on attention in the test set of the category Mine Block. We can observe that the introduction of the new attention mechanism significantly accelerates the model's performance improvement during training and helps the model reach a higher performance ceiling. This demonstrates the necessity of the dual-path architecture, which enables the model to quickly encode the relationship between video instructions and current environmental observations through the introduction of intention-aware attention. Consequently, the model can generalize the visual feature matching patterns to the test set environments, enhancing its overall generalization capability in task execution.

### 4.4 ABLATION ON BASE POLICY

To emphasize the effectiveness of our structure, we used the same reinforcement learning method to fine-tune STEVE1 on the Mine Block task using both text and video instructions. The training results, as the training iterations change, are shown in Figure 7.

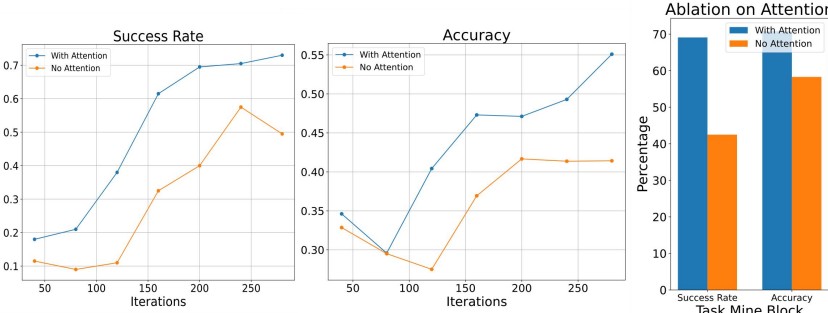

Figure 5: **Ablation on attention:** The ablation experiment on whether to introduce the new pathway by cross attention in the MineBlock category task. This figure shows the training iteration count on the x-axis and the model's success rate and accuracy on the test set on the y-axis. It can be seen that the dual-pathway information transfer structure with attention significantly improves the overall generalization performance of the agent.

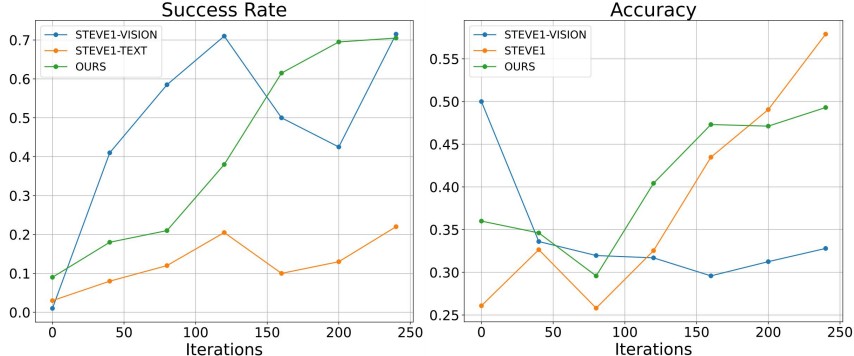

Figure 6: **Ablation on Base Policy** We online finetuned STEVE1 and STEVEv(STEVE1 with vision instruction in Lifshitz et al. (2023)) as our baseline. Compared to STEVE1, which conservatively focuses on mining items it is confident about, and STEVEv, which aims to mine as many items as possible, our agent excels in both success rate and accuracy, striking a good balance. This demonstrates the strong generalization capability endowed by our dual-pathway architecture.

## 5 RELATED WORKS

**Multi-task Reinforcement Learning**    Multi-task Reinforcement Learning (MTRL) is a reinforcement learning method that aims to enable an agent to learn and optimize across multiple tasks. Unlike single-task reinforcement learning, MTRL requires the model to handle multiple tasks simultaneously, to allow the model to share knowledge between tasks, thereby improving learning efficiency and generalization capabilities. Most of the previous works mainly focus on in-distribution tasks Guo et al. (2020). MTRL struggles with many problems such as overfitting and gradient conflict. which are similar to the challenges faced when attempting to share features between the actor and critic in actor-critic methodsCobbe et al. (2021). To solve this issue, many works focus on optimization was proposed. Some other works focusing on RL generalization have been proposed.

The capability of RL to perform few-shot generalization across environments has been validated on Atari by Taiga et al. (2023).Works like Zhou et al. (2024) use the images as the instruction, which is similar to our work, however, it lacks semantic feature and losing the ability of base agent.

**Developing Agents in Minecraft**   Minecraft is an open-ended, voxel-based game that provides a complex environment for a wide range of tasks, making it both a challenging and inspiring plat-form for developing and benchmarking decision-making agents. In recent years, it has attracted increasing attention from researchers. Platforms like MineRL (Guss et al., 2019) and MineDojo (Fan et al., 2022) offer customizable environments and large-scale datasets, facilitating the training and evaluation of agents. One stream of work (Wang et al., 2023a;c;b) focuses on developing plan-ners that utilize large language models to generate high-level actions, and rely on separately trained controllers or leveraging game mechanics to carry out these actions. In contrast, another stream of work, more closely related to our approach, focuses on developing low-level controllers that di-rectly interact with the environment. VPT (Baker et al., 2022) learns an inverse dynamics model to infer mouse and keyboard actions from YouTube videos and trains a behavior cloning model on a web-scale dataset. They further fine-tuned the model with KL-constrained reinforcement learning, achieving impressive results on the diamond acquisition task. However, their method is limited to solving only a single task. In this work, we apply the same KL-constrained reinforcement learning technique to fine-tune a model pre-trained on a large dataset, but with a focus on the multi-task setting. Fan et al. (2022) finetunes a pre-trained contrastive video-language model for Minecraft, which can be used to embed text instruction and generate reward signals. While it trains a language-conditioned policy, its reinforcement learning approach isn't built on large-scale pre-training, and its generalization ability is not tested. Lifshitz et al. (2023) combines MineCLIP and VPT to train a text-conditioned policy, while Cai et al. (2023) proposes a self-supervised approach for training a video-conditioned policy. However, both approaches remain fully offline without interaction with the environment, which limits their performance.

# 6 CONCLUSIONS

In conclusion, we recognize that text instruction requires an expensive and complex alignment pro-cess. Therefore, we propose that video instruction has the potential to enhance the generalization capability of online algorithms. During this process, we discovered that when the number of tasks is limited, the latent features encoded by video instruction are prone to overfitting to the training task space, making it difficult to generalize to unseen tasks. To address this, we introduce a dual-stream encoder-decoder architecture:

1. It leverages latent features pre-trained on unlabeled data to provide high-level behavioral information, thereby improving the initial exploration process of reinforcement learning.

2. It utilizes an attention-based visual communication module to provide more fine-grained visual concepts for specific tasks, allowing the policy to learn how to combine the environ-ment with the instruction, thus enhancing generalization.

Our method outperforms nearly all other common agents in Minecraft environments on the gener-alization benchmark we designed for three types of tasks. This ability to train on small sets and zero-shot generalize to open-ended tasks lays the foundation for applying online algorithms in de-veloping generalist agents in open-world environments.

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

## A MINECRAFT ENVIRONMENT

Minecraft is a popular open-world sandbox game that allows players to interact with and modify a procedurally generated world. The game offers an almost limitless number of tasks and presents challenges similar to those found in the real world, making it an ideal testbed for AI research. A generalist agent operating in Minecraft must handle partial observability to navigate the complex 3D environment, deal with the dynamic and sparse-reward nature of the game, engage in long-term planning for multi-step tasks like resource gathering and crafting, and generalize to varying weather conditions and changing landscapes. Additionally, as a popular game, Minecraft provides an abundance of training data online, including gameplay videos on YouTube and text-based tutorials

Table 2: **Training HyperParameters**

| HyperParams | Value | HyperParams | Value |
|---|---|---|---|
| Total Iterations | 60 | GAE lambda | 0.95 |
| Learning Rate | 2e-5 | PPO CLIP | 0.1 |
| Schedular | Linear | Policy Loss Weight | 1.0 |
| Optimizer | Adam | Value Loss Weight | 0.5 |
| Adam eps | 1e-8 | KL Loss Weight | 0.2 |
| Training GPU num | 2 | KL Loss Decay | 0.995 |
| Batch Size per GPU | 1 | epochs per iteration | 1 |
| Batches per Iteration | 200 | Context length | 128 |
| gradient accumulation | 10 | Reward Discount | 0.999 |

in forums and wikis. This vast amount of unstructured, unlabeled data makes Minecraft a valuable environment for researching how to leverage such large-scale datasets for AI training.

In this work, we use Minecraft as a testbed for multi-task reinforcement learning with large-scale pretraining. Our environment setup aligns with the one used in VPT, where agents interact with the game in the same way as human labelers: they receive only RGB game screenshots as visual input and perform low-level actions through mouse and keyboard at 20Hz. No high-level observations, such as voxel data, are provided to the agents.

## B  TRAINING SETUP

We use MineRL to facilitate interaction with the environment. In all tasks, the agent spawns on a plain.

### B.1  TRAINING ENVIRONMENT

In the Mine Block task, we randomly placed three distinct blocks within a 2x2 to 11x11 range centered around the agent, with one of the blocks corresponding to the block in the instruction video. The agent receives a reward of 10 upon mining the correct block while mining either of the other two distractor blocks results in a penalty of 4.

### B.2  INSTRUCTION COLLECTION

To obtain the instruction videos, we manually recorded two types of tasks: crafting items and killing entities. In the manually recorded environment, there were no distractors, and the target items were placed three blocks ahead of the agent's spawn point, with other settings consistent with the training environment. In the videos recorded by human players, the success rate of achieving the goal was 100%. For the MineBlock task, we used Groot to follow a fixed trajectory recorded by a human in the recording environment, then generated 10 additional trajectories in bulk within the same environment. The success rate was not 100%, but the model's performance was sufficient to convey both action and visual information. Under this recording setup, we recorded 5-10 instruction videos for each task within each category. During actual training, the agent randomly selects one of these trajectories corresponding to the task as the instruction video.

### B.3  HYPERPARAMETERS

Here is our hyperparameters for training.

In the ablation of the base model, we still face problems: At the beginning of training, due to the presence of penalties, both STEVE and STEVEv models struggle to explore positive rewards. Therefore, in these experiments, we first took the absolute value of the negative rewards for the initial 120 iterations, changing -4 to +4 to encourage the agent's exploratory behavior. After 120 iterations, the agent obtained from the 120th iteration served as the initial checkpoint, and training continued using the original parameters. However, to ensure that the KL divergence remains within

a reasonable range, the reference model must be capable of performing various behaviors, making the checkpoint from the 120th iteration unsuitable. Hence, in the related experiments, the reference model was always kept as the base model from the very start of training.

