# OpenReview forum: "Training Open-ended Policies to follow Video-prompt Instructions with Reinforcement Learning"
_ICLR.cc/2025/Conference — ICLR 2025 Conference Withdrawn Submission_

### Official Review · Reviewer_ZNbS · 2024-10-22

**Soundness:** 2
**Presentation:** 1
**Contribution:** 1
**Rating:** 3
**Confidence:** 4

**Summary:**

The paper proposes a method to train agents with reinforcement learning to solve tasks in Minecraft. It extends previous work based on video-conditioned agents (the GROOT method), adding reinforcement learning training and architectural modifications to better handle generalization over tasks. The method is evaluated on an ad-hoc suite of tasks.

**Strengths:**

- The goal of building agents able to execute tasks zero-shot, without any task-specific training, is important and somewhat under-explored for agents trained with reinforcement learning.
- The architectural modifications proposed in the paper are sensible. Giving the action decoder a more direct access to the low-level details of the input can potentially better inform a policy.

**Weaknesses:**

- I found the premise of the paper, repeated multiple times, to be highly misleading. While using video instruction might be more convenient in some circumstances, the value of text instructions mostly resides on how it is for a user to input them into the system. This should be taken into account as a premise to the work and cannot be ignored.
- Generally, the paper is very unclear. It is quite difficult to understand what the method exactly is, what the exact training procedure is, and how the evaluation was carried out.
- There are no error bars for any of the plots, and especially given the variance of online reinforcement learning algorithms, they are required to understand the validity of the empirical results.
- The paper is full of editorial mistakes (e.g., no space before in-line citations, missing periods, typos, placeholder captions [Figure 3]). I suggest the authors to review their manuscript to fix them.

**Questions:**

- How was the reinforcement learning training done? How were the rewards designed?
- How is the "accuracy" exactly defined?
- How many runs did you use?
- What is the variance across different runs?

---

### Official Review · Reviewer_1hDH · 2024-10-31

**Soundness:** 3
**Presentation:** 1
**Contribution:** 2
**Rating:** 5
**Confidence:** 3

**Summary:**

The paper explores applying video instructions to open-ended policies for better generalization. However, a naive method of training the policy results in poor generalization due to the latent space mainly capturing semantic features, while neglecting the fine-grained visual features. To mitigate this, the paper proposes intention aware attention architecture where the model attends to both visual and semantic features through cross-attention. The model is trained with online RL method. Results show that the proposed model significantly outperforms baseline models on Minecraft tasks.

**Strengths:**

- The proposed method significantly outperforms baseline models on three minecraft tasks.
- The paper evaluates the performance on both success and accuracy metric, making the evaluation setup solid and fine-grained.

**Weaknesses:**

- The paper contains many grammar errors and typos which significantly hinders the readability of the paper. For example, line351-353 and line358-360 are identically the same. / Table 1 (Generalizaiton -> Generalization) / Caption of Figure 3 / many capitalization errors (However, In -> However, in (line 149), VPT work, In such -> VPT work, in such (line 209)) / The numberings of the figures in the main text is not aligned with the paper text (line 420 Figure 4.3 -> Figure 5 / line 431 Figure 7 -> Figure 6).
- The contribution of this paper is unclear. Applying video instruction instead of text instruction is already explored in the GROOT paper. Also, the problem that the "latent z tends to encode actions" can be only applied for the GROOT architecture, which limits the applicability of the paper. Overall, the contribution of the paper is limited compared to GROOT.
- The experiments mainly focus on simple short-horizon tasks, while GROOT also shows results on long-horizon task (obtain diamond).

**Questions:**

- How are instruction videos defined during unseen task inference? Are the instruction videos different for different entities? If this is the case, the player should provide instruction video demonstration for every task that interacts with a specific entity, which limits the applicability due to overhead during evaluation. On the other hand, language instructions are much more simple and does not require any prerequisites during at test time.
- What is the effect of online RL used in this paper? Would the performance also enhance when applying imitation learning to intention aware attention architecture?

---

### Official Review · Reviewer_2JuP · 2024-11-01

**Soundness:** 2
**Presentation:** 1
**Contribution:** 2
**Rating:** 3
**Confidence:** 3

**Summary:**

This paper proposes to train agents with video demonstrations, rather than textual instructions, on Minecraft. Despite some potentially interesting results, the presentation of the paper is very confusing which makes it difficult to evaluate, and it does not meet the bar for publication at ICLR. Therefore, my recommendation is reject. I have included a detailed list of questions which hopefully can help improve the presentation.

**Strengths:**

- Overall, it’s good to see work on Minecraft, since it is one of the more challenging decision-making environments available.
- The idea of using video instructions is reasonable, since giving examples of the desired behavior is often easier than specifying a reward function.

**Weaknesses:**

The presentation of the paper is very confusing with many important details missing, in addition to formatting issues and typos. As such, it is difficult to understand how exactly the method is working and if the comparisons make sense and are fair. It's also unclear what data they are using, and if the baselines all have access to the same data. Finally, standard methodological criteria such as including error bars are not met. Please see my detailed list of question below.

**Questions:**

Major:

- How is the goal-specific reward function $r^g_t$ defined in Section 2.1? Also, the goal space should appear somewhere in the definition of the reward function $R: S \times A \rightarrow \mathbb{R}$, right? Also, $t$ and $g$ should be italicized on the last line of Section 2.1.
- Section 3.2 is redundant, it just specifies the standard PPO update rules.
- On the other hand, the reward function that is used is not defined anywhere. Other methods like MineCLIP use the similarity between the embeddings of the image observations and textual instructions. But here, the paper says it is using video instructions. Is some kind of analogous reward defined, consisting of the similarity between instruction embeddings and the observation embeddings?
- More generally, the proposed algorithm should be clearly described someplace. Please add some pseudocode or a detailed diagram (Figure 2 does not show any rewards, and my understanding is that this is not the proposed model but a baseline). It would be good to have the learning objetives spelled out somewhere (my understanding is the the reward function is the sum of multiple goal-conditioned rewards?)
- What is the demonstration data used? How many trajectories/frames does it consist of? All I see in the paper is the sentence “We recorded each task’s instruction videos manually and generated them using the model, showcasing the player successfully performing the correct actions on the target objects.” Do you also have textual instructions for all these demonstrations? How are you training the baselines to ensure they all have access to the same data for a consistent comparison?

Minor:
- Section 2 title: “Preliminary” should be “Preliminaries”.
 Line 047: “In this work. We draw...” is not a proper sentence.
- References are not properly formatted, see line 035 for example.
- Figure 3’s caption says “Draft” and parentheses are not matched.
- Line 379: “Generalization” is misspelled.
- The list of typos above is not exhaustive, there are others throughout the paper. Please check the writing carefully for the next revision, with the help of an LLM if needs be.

---

### Official Review · Reviewer_qU6u · 2024-11-02

**Soundness:** 3
**Presentation:** 2
**Contribution:** 3
**Rating:** 3
**Confidence:** 3

**Summary:**

This paper builds on the GROOT encoder-decoder architecture by proposing novel attention layers to extend a semantic and visual dual-pathway structure within the base model. This enhancement significantly boosts the generalization capabilities of the video instruction model for online reinforcement learning (RL). The proposed method demonstrates substantial improvements in zero-shot generalization, achieving success rates of 69.1%, 16.0%, and 67.7% across three categories of unseen tasks in the Minecraft environment.

**Strengths:**

1. This work addresses a critical challenge in online reinforcement learning (RL): zero-shot generalization.
2. The proposed dual-pathway model architecture offers more fine-grained visual concepts as instructional signals, thereby enhancing the generalization capabilities of the base model.
3. Some quantitative results validate the effectiveness of the dual-pathway attention mechanism in enhancing the generalization capabilities of instruction-tuning models.

**Weaknesses:**

$\textbf{Lack of Clarity in the experimental section:}$

1. The clarity of the ablation section needs improvement. For instance, the authors should clarify the settings for "with Attention" and "No Attention" in the ablation study on intention-aware attention.

2. In the ablation analysis of intention-aware attention, it appears that "No Attention" is equivalent to the base policy model GROOT. If this is not the case, has the proposed method refactorized the projection of latent z to address the limited feature dimension for enhancing generalization?

3. The ablation section on the base policy contains some confusing aspects regarding the baseline policy. The paper states, "Given GROOT’s outstanding ability to follow video instructions, we chose it as our base policy" (line 175). It seems that GROOT is intended as the comparison baseline in the ablation for the base policy. However, the analysis compares the dual-pathway architecture to STEVE1 and STEVEv, which is not a fair comparison since the dual-pathway architecture is not the only controlled variable between STEVE and the proposed model.

$\textbf{Additional experimental analysis:}$
I recommend including an ablation study on the KL term "between the current policy and the original policy," as this would further validate the effectiveness of the KL constraint for the agent.

While the proposed method shows promising direction to enhancing the generalization capabilities of video instruction-based online RL, the paper lacks clarity regarding key components, such as the policy model architecture. This causes limitations in both the reproducibility and readability of the proposed method. Furthermore, some experimental settings are not normative to ablation study, or lack a detailed analysis of each design (e.g., objective function). The above reasons lead me to conclude that the current version of the manuscript is not suitable for publication; therefore, my rating tends to be below the accepted threshold.

**Questions:**

Please refer to the Weaknesses section.

---

### Official Review · Reviewer_eoLh · 2024-11-02

**Soundness:** 3
**Presentation:** 1
**Contribution:** 1
**Rating:** 3
**Confidence:** 3

**Summary:**

This paper explores the use of video demonstrations as a method for improving the generalization ability of agents in open-ended multi-task learning environments. The authors argue that by providing agents with video prompts, they can learn to perform a wider range of tasks, even those not explicitly seen during training. The paper presents experiments in Minecraft to demonstrate the effectiveness of their approach.

**Strengths:**

- Introduces a novel dual-pathway architecture (semantic + visual) that appears to meaningfully improve within-category generalization performance
- Identifies and addresses a specific failure mode in encoder-decoder models where the latent space overfits to a "task vocabulary" during online finetuning

These technical contributions represent meaningful progress in improving generalization within structured task categories if one has access to video prompts of the desired task.

**Weaknesses:**

By providing the agent with a video of the task it needs to perform, the agent is given a step-by-step demonstration of what is expected, and must only learn to parse the action sequence contained in the human-provided video prompt at inference time.

Doing this is not entirely without merit: this may well be a non-trivial problem to solve if the prompt is out of distribution with respect to the training data.

However, I am not convinced that solving this problem is at all relevant to open-endedness: the proposed agent relies on being provided a concrete sequence of actions to take (contained in the video), but that is precisely what one hopes that agents can discover autonomously.

In other words, access to the video prompts that would be required in this set up does not seem feasible in practice, so I fail to understand the real-world relevance of this technical contribution.

The reason language prompts are prevalent in the literature is because natural language allows conveying arbitrarily abstract ideas, so requiring natural language prompting is a less strict limitation, as long as the agent can interpret abstract goals. (Though this is still debatable, as ultimately open-endedness is about creating new goals.)

In order to strengthen these results, the authors would have to argue how obtaining these video prompts without human intervention would happen, and how this would still be relevant despite the limitations of imitation learning. For example, in a multi-agent setting, one could show that one can build an agent that wanders observing other players, scores what interesting new observed skills to learn, and utilizes the proposed methodology to internalize these skills.

I have a few other qualms with the methodology and soundness of the claims, but these seem second-order considerations given the above.

I may well have misunderstood the motivation and relevance of the work. If this is the case, perhaps the authors can clarify the problem statement and any other relevant details to highlight the relevance of the contributions to the field of open-ended reinforcement learning, and I would be very happy to re-assess their technical contributions more fairly.

I also encourage the authors to improve their writing and organization of the manuscript.

**Questions:**

Given that the approach relies on video prompts, can you clarify how this setup maintains relevance to open-ended task generalization, considering that agents receive a step-by-step guide?

---

### Note · Authors · 2024-11-15

I have read and agree with the venue's withdrawal policy on behalf of myself and my co-authors.